# Attacking quantum key distribution by light injection via ventilation openings

**Juan Carlos Garcia-Escartin** [1] *, **Shihan Sajeed** [2,3,4,5], **Vadim Makarov** [4,6,7,8]

**1** Dpto. Teoría de la Señal y Comunicaciones e Ing. Telemática, Universidad de Valladolid, Valladolid, Spain, **2** Institute for Quantum Computing, University of Waterloo, Waterloo, ON, Canada, **3** Department of Electrical and Computer Engineering, University of Waterloo, Waterloo, ON, Canada, **4** Department of Physics and Astronomy, University of Waterloo, Waterloo, ON, Canada, **5** Department of Electrical and Computer Engineering, University of Toronto, Toronto, Canada, **6** Russian Quantum Center, Skolkovo, Moscow, Russia, **7** Shanghai Branch, National Laboratory for Physical Sciences at Microscale and CAS Center for Excellence in Quantum Information, University of Science and Technology of China, Shanghai, People's Republic of China, **8** NTI Center for Quantum Communications, National University of Science and Technology MISiS, Moscow, Russia

* juagar@tel.uva.es

**Data Availability Statement:** All relevant data are within the manuscript and its Supporting Information files. The supplementary data are also uploaded to the Universidad de Valladolid

## Abstract

Quantum cryptography promises security based on the laws of physics with proofs of security against attackers of unlimited computational power. However, deviations from the original assumptions allow quantum hackers to compromise the system. We present a side channel attack that takes advantage of ventilation holes in optical devices to inject additional photons that can leak information about the secret key. We experimentally demonstrate light injection on an ID Quantique Clavis2 quantum key distribution platform and show that this may help an attacker to learn information about the secret key. We then apply the same technique to a prototype quantum random number generator and show that its output is biased by injected light. This shows that light injection is a potential security risk that should be addressed during the design of quantum information processing devices.

## 1 Introduction

In modern computer networks, users need fast and secure channels. Key distribution protocols based on computational assumptions, such as the RSA cryptosystem [1], enable the initial key exchange that these channels require. Quantum key distribution (QKD) protocols [2, 3], like BB84 [4] or Ekert [5] protocols, and their research [6–9] and commercial [10] implementations offer a physics-based alternative.

For ideal systems, the laws of quantum mechanics guarantee that any existing eavesdropper is detected [11–16], but practical implementations can deviate from the original assumptions. There are multiple experimental demonstrations of quantum hacking that exploit imperfections in the detectors [17–26] or problems with state preparation [27, 28], among others.

One of the assumptions of QKD is that the equipment is sealed from the outside, but this is not necessarily the case. In this Article, we show a new potential attack vector due to ventilation holes. More specifically, we show how an attacker can expose unintentional light paths

repository and publicly accessible via the following URL: http://uvadoc.uva.es/handle/10324/41329.

**Funding:** This work was supported by Junta de Castilla y León (project VA296P18), MINECO/FEDER, UE (project TEC2015-69665-R), Junta de Castilla y León (project VA089U16), Movilidad Investigadores UVa-Banco Santander 2015, Industry Canada, CFI, NSERC (programs Discovery and CryptoWorks21), Ontario MRI, US Office of Naval Research, and the Ministry of Education and Science of Russia (program NTI center for quantum communications).

**Competing interests:** The authors have declared that no competing interests exist.

into the interior of quantum systems to inject additional photons that break the original assumption that the pulses are either single photons or weak coherent states with a controlled mean photon number.

The attack can be extended to quantum random number generators (QRNG), which are devices that take advantage of the inherent randomness in quantum mechanics to produce random bit sequences. Many commercial models and prototypes are based on measuring the quantum states of light [29] and must be protected from external photons.

The Article is structured as follows. First, we introduce optical attacks on security systems in Sec. 2. Then, we address the feasibility of ventilation hole attacks on quantum optical devices and show experimental examples of attacks on a QKD system and a QRNG in Sec. 3 and Sec. 4, respectively. We discuss potential eavesdropping risks due to light injection attacks and conclude in Sec. 5.

## 2 The optical side channel and light injection attacks

Attacks with light are, in many ways, related to electromagnetic attacks [30–32], but, in classical electronic systems, light offers fewer possibilities for side channel and injection attacks than methods that use electromagnetic radiation up to the GHz range. Most electronic systems are only slightly sensitive to light, if at all.

There are, however, a few classical examples that help to understand the relevance of light injection attacks. In semiconductor cryptographic chips, the photons emitted from the transistors that are active during encryption can be exploited to deduce the secret key stored in the device [33, 34]. These attacks require access to the chip and invasive methods like decapsulation. Similarly, if we have the chip in our possession, laser light injection can induce faults during encryption that reveal the secret keys [35].

Light from the devices can also give information to attackers that cannot reach the system directly. Most electronic devices use light-emitting diodes (LEDs) to signal normal operation and for quick visual diagnosis. Many network cards have an LED that shines when data is sent or received and, depending on the concrete circuit design, the pattern of the light can follow the bit streams and leak information about the transmitted or received data [36]. These LEDs are, by design, well visible and servers tend to be in plain sight, usually behind glass doors, back to back to unknown equipment, or even by windows (see Fig 1).

These optical paths also open a backdoor for optical injection attacks where an optical signal alters the normal operation of the device. External light has already been behind some spontaneous failures in photosensitive components. One such event is the accidental activation of the halon fire suppression system in the Haddam Neck nuclear power plant in 1997 when a camera flash affected the contents of an exposed EPROM memory inside a cabinet [38]. More recently, a similar camera-shy behaviour has been noticed in the popular Raspberry Pi single-board computer. The Raspberry Pi 2, Model B, version 1.1, was noticed to turn off due to an exposed component of the power supply, a phenomenon which has been colourfully dubbed the "xenon death flash" [39]. The photoelectric effect in silicon is behind these two examples and they share a simple solution: covering the offending component, which was, anyway, designed to operate under a cover.

There have also been planned attacks on devices designed to be secure, like slot machines. The payout mechanism of some models counts the returned coins with a light sensor. Police have found some cheaters blinding the optical detector inside the machine with a "light wand", a simple light source that could be introduced in the coin slot. The blinded stop mechanism failed and the coin reservoir was emptied whenever there was a prize, no matter how small [40].

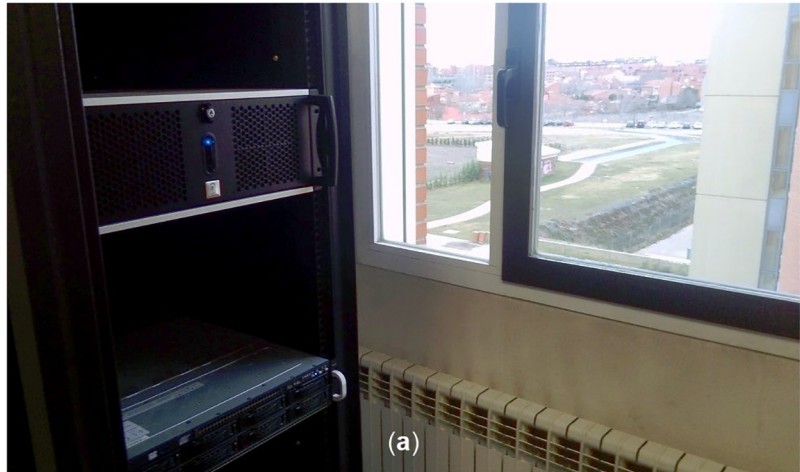

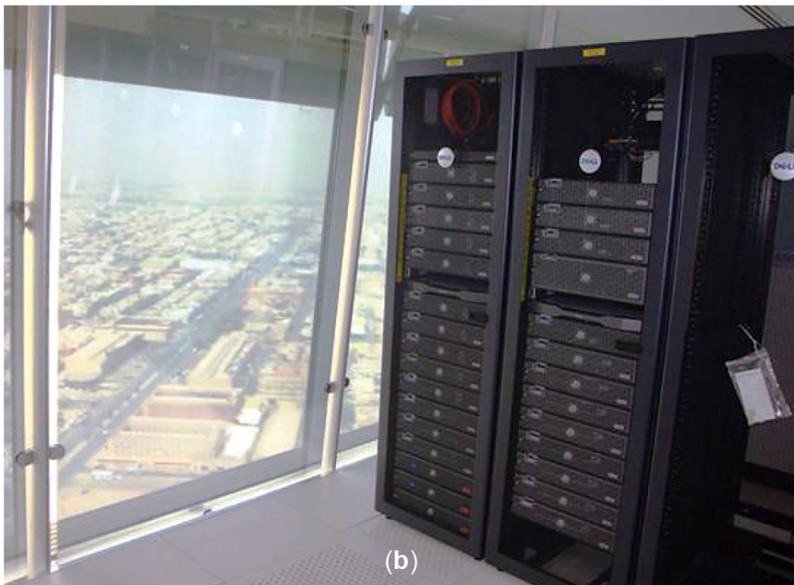

**Fig 1. Server racks with a view to the outside.** (a) A picture taken at author's (J.C.G.-E.'s) own university. (b) Computer server room in Al-Faisilya Tower, Riyadh, Saudi Arabia [37].

These few examples notwithstanding, light injection attacks are a small concern in classical systems. In electronic systems, there are few components that can be affected by light and the systems that primarily use optical signals, like the optical fiber backbone that carries most internet traffic, are well isolated. The optical signals have relatively strong optical power levels and external light couples very weakly to the inside of the optical fiber and the other optical components. An attacker would need to inject a signal with a large optical power before achieving any effect. In contrast, most quantum systems work with only a few photons and need to address light injection attacks explicitly.

Here, we take advantage of the fact that most electronic and optical devices need ventilation to take away the excess heat produced during operation. QKD systems include electronic processing elements and optical devices like lasers, all of which require some form of heat removal for a correct, stable operation. In particular, at the infrared telecommunication wave-lengths that are required to take full advantage of the existing optical fiber technology and

infrastructure, single-photon detectors have notoriously poor performance unless properly cooled [41, 42]. The devices that are used in QKD must have a proper thermal design that most likely will involve ventilation holes to circulate the air in and out of the machine.

Any ventilation hole that allows an optical path to an uncovered optical device is a potential threat to security. External light can enter unjacketed or poorly covered fiber. While, under normal circumstances, the coupling is too weak to be noticed, an attacker could introduce a few additional photons that can make a huge difference in quantum applications.

## 3 Case study: Photon injection into a QKD system

Here we show that photon injection is a plausible attack vector for QKD. The attack has its limitations, but it is a potential threat and must be taken into account. In this Section we describe the system under study, introduce the basic attack, and present and analyse the results of a proof-of-principle experiment.

### 3.1 Device under study: Plug-and-play QKD

For our proof-of-concept we study the Clavis2 QKD platform, which is based on a plug-and-play scheme [23, 43]. Clavis2 was designed by ID Quantique for research and development applications, and is now discontinued [44]. Fig 2 shows the basic configuration of the involved devices. Two sides, Alice and Bob, establish a secret key using an optical fiber link. Bob sends to Alice a sequence of pulses grouped in pairs at classical power levels through the optical fiber channel. Alice measures the power of the classical pulses she receives [23], and sets a variable optical attenuator (VOA) to guarantee that the signals that get out have a proper mean photon number (typically less than one photon). Having quantum signals is what makes the whole system secure.

Alice chooses a phase from $\left\{0, \frac{\pi}{2}, \pi, \frac{3\pi}{2}\right\}$ and encodes it in the second half of the pulse pair using a phase modulator (PM), which is located after a delay line (DL) that helps to avoid problems with Rayleigh backscattering [43, 45]. Alice's setup is completed with a Faraday mirror (FM) that compensates for polarization asymmetries in the channel. The pulses then go back through the DL, are attenuated again and cross a 10:90 fiber coupler (C) before leaving Alice.

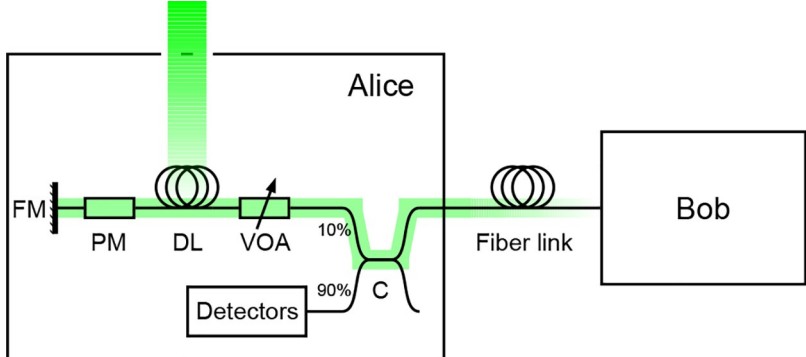

**Fig 2. Plug-and-play QKD device under attack.** Alice's side includes a 90:10 optical coupler (C) that diverts part of the light for monitoring to a series of classical detectors, and the quantum part with a variable attenuator (VOA), delay line (DL), phase modulator (PM), and Faraday mirror (FM). Alice uses the PM to introduce a secret phase and makes sure that the total attenuation guarantees that the mean photon number per pulse going back to the fiber channel is less than the value prescribed by the protocol. The coupling path in our attack is shown in green (gray): a ventilation hole in Alice's case gives light an access to the fiber spool of the DL. From there, the injected photons can get to the phase modulator and carry unwanted information to the outside fiber channel.

When Bob receives the single-photon pulses, he chooses a random basis for his measurement. Depending on this choice, he can perfectly distinguish either between Alice's states {0, $\pi$} or $\left\{\frac{\pi}{2}, \frac{3\pi}{2}\right\}$. If the random selections of Alice and Bob are matched, they know Bob's measurement will show the random bit from Alice. An eavesdropper, Eve, monitoring the channel cannot learn these values without introducing errors and thus being detected.

Our attack targets the delay-line fiber spool, which in our device under study consists of 10.53 km of optical fiber. In plug-and-play QKD, the photons are sent in trains of pulses and the timing of the trains and the length of the DL are chosen so that forward and backward travelling light only meet inside the fiber spool [43, 45].

## 3.2 Theory of attack

*Principle*: The attack uses a ventilation hole on Alice's side that allows an eavesdropper, Eve, to couple extra light into the delay-line fiber spool. In order to minimize bend losses, the fiber is wound around a spool with an internal diameter of around 16 cm. The side of the spool is open and the fiber in primary coating is exposed to the outside with the only protection of a thin plastic wrapping. The spool is placed close to a ventilation opening (see Fig 3). This location aligns well with standards that require electronic parts not be reachable from the outside [46], but it allows us to perform light injection attacks.

In our QKD fiber system, the light travels through monomode optical fiber, where different mechanisms introduce losses. We are mostly interested in reversible loss mechanisms which couple some of the photons from the guided modes of the fiber into radiated modes that leak to the outside. The two most important reversible loss mechanisms are Rayleigh scattering and bend losses. Rayleigh scattering is the dominant cause of loss in silica fibers at infrared wavelengths [47]. Additionally, bends in optical fiber can lead to losses [48–52]. This weak coupling to the outside is reversible and light coming from the outside can couple to the core and remain inside the fiber. This principle has been used before to design test tools for optical fiber networks that inject light through small bends without the need for a connector or a splice [53, 54].

In our attack, light is injected into the fiber using the reverse from these processes. Measurements with an equivalent system with a spool of fiber removed from the case suggest that light injection comes from a combination of Rayleigh scattering and bending effects. Comparing the coupling of light at 1310 and 1550 nm, we see that more photons enter the fiber at the lower 1310 nm wavelength, where Rayleigh scattering is stronger for silica fibers. On the other hand, for both wavelengths, the number of injected photons varied with the input angle of a collimated laser beam with respect to the spool (increasing the coupling for almost tangential incidence). This suggests the geometry of the fiber also plays a role in the coupling and there is a component related to direct coupling at bends.

*Attack setup*: The injected light passes both inwards—towards the phase modulator—and outwards—towards the channel. The photons coupled towards the channel do not carry any useful information. We want to send the light towards the phase modulator so that it will carry the secret phase information from Alice.

The real situation is complex and includes the effect of the plastic wrapping, the fan, and multiple other details, but the heuristic of directing the light close to the tangent to the curved fiber gave the highest coupling to the spool. Not all the parts of the spool can be reached through the ventilation hole. If possible, the light should enter the spool close to the modulator output, which minimizes the total loss.

In our experiment, the divergence of the beam was not critical. A fiber laser was connected to a fiber collimator a few centimeters from the ventilation hole giving a loosely focused spot

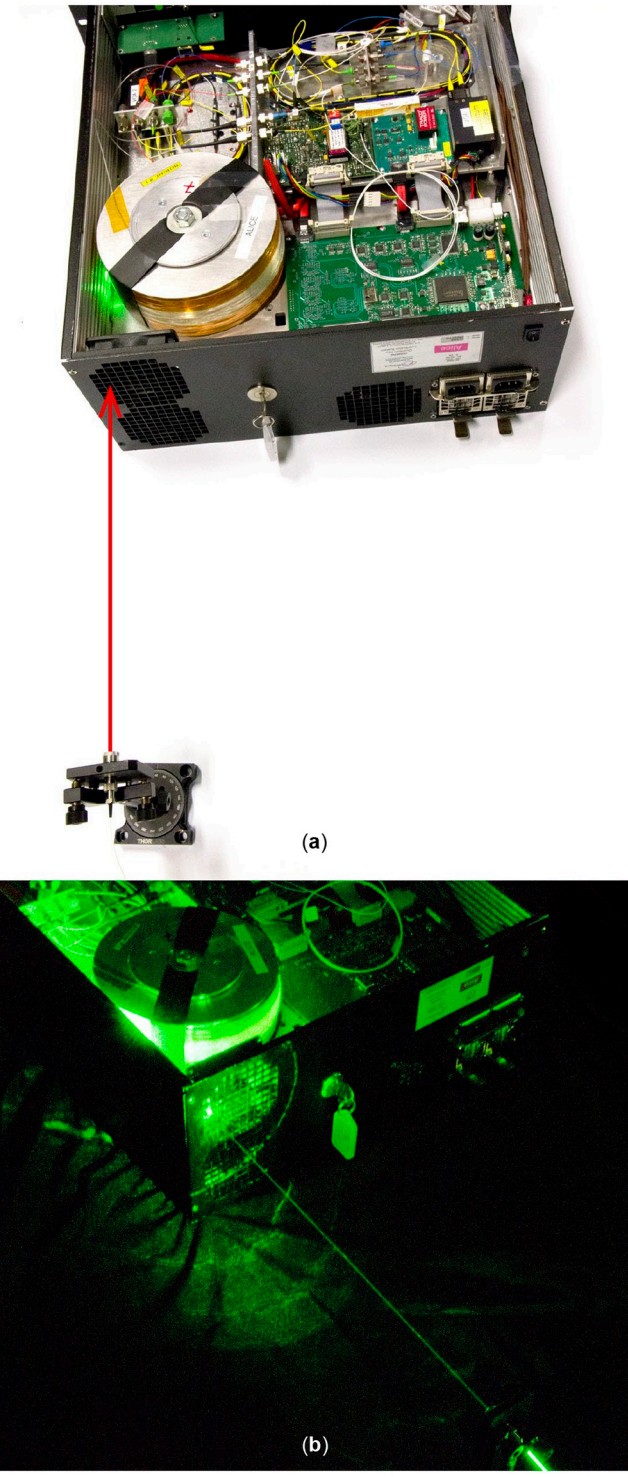

**Fig 3. Experimental setup for QKD system Clavis2.** (a) A collimated laser light beam (denoted by an arrow) from a fiber collimator passes the ventilation hole and couples to the delay line. We have removed a solid metal cover from the system to show its internals. (b) A picture taken in the dark shows how the light from the beam couples to the fiber spool (for this picture we used a visible green laser instead of 1536 nm).

on the fiber spool (of a couple millimeters diameter), illuminating multiple fibers in it. The beam properties can only be controlled up to the trellis protecting the ventilation hole. In the path to the fiber, the beam is modified, particularly by the random rugosity of the plastic that surrounds the fiber spool. Even if the shape of the spot reaching the fiber changed, during the experiment the injected number of photons did not show any significant variation when the laser hit different parts of the spool. The angle of the beam seemed to be the most relevant parameter when there was a clear line of sight.

The injected photons arriving at the PM—when it is active—collect the phase information and get reflected by the Faraday mirror. They then pass through the lossy components in Alice and come out into the channel. Eve can then measure them to extract the secret encoded value. In this way, this attack is equivalent to a Trojan-horse attack [55, 56], but it cannot be detected by monitoring the input fiber. Alice's side in Fig 2 shows a schematic representation of our attack.

We did not study the temporal characteristics of the coupling. For simplicity we used a continuous-wave (c.w.) laser. However, the attacker can adapt the time profile of her light to maximize the number of photons that reach the active PM. A realistic attack will be in between this ideal limit and our c.w. approach. The trade-off between timing precision and total power is discussed in sec. 3.4.

*Attack efficiency*: Eve can compromise the system if she is able to deduce Alice's phase settings. If Eve uses pulses longer than the time bin the PM is active, each output photon is equivalent to the time bin qubits used in the QKD protocol. The part of the light reaching the inactive modulator serves as a phase reference. Experimentally, a simple alternative would be using a homodyne detection setup (see [57] for an example of the method in a Trojan-horse attack on QKD).

We will consider attacks where Eve uses all of the output light (injected and legitimate photons). If Eve injects photons at a slightly different wavelength than the photons from Bob, she could also tell apart the injected photons from the rest with a wavelength demultiplexer, if needed. Given the low coupling we found during the experiments, Eve's best strategy seems to be using also the photons in the legitimate channel.

The success of Eve's attacks depends on how many photons she can get at the system output. While we can experimentally measure injection efficiency into the DL, the photons suffer additional losses inside Bob before they reach the channel. We discuss the latter below.

We first consider normal QKD operation. Let's assume $\mu_A$ ($\mu_B$) is the mean photon number per pulse coming out of Alice (Bob), $t$ is the channel transmission, $\mu_A = 2(\mu_{coup} + \mu_{VOA} + \mu_{spool} + \mu_{PM})$ is the total round-trip loss inside Alice with $\alpha_{coup} = 10.8$ dB, $\alpha_{PM} = 4.2$ dB $\alpha_{spool} = 4.6$ dB $\alpha_{VOA}$ being the losses measured in the coupler, phase modulator, fiber spool and VOA respectively. We assume the Faraday mirror introduces a negligible loss. For these values

$$\alpha_A = 2\alpha_{VOA} + 39.2 \text{ dB}. \tag{1}$$

The mean photon numbers $\mu_A$ and $\mu_B$ are related as

$$\mu_A = \mu_B t 10^{-\alpha_A/10}. \tag{2}$$

The mean photon number $\mu_B$ can be determined from experimental measurements. In Ref. [23], the energy of the pulse coming out of Bob was measured to be $E_\mu$ 73 fJ; which leads to $\mu_B = E_\mu \lambda/(hc) = 5.69 \times 10^5$.

We assume that the system sets the value of the VOA in such a way that optimizes $\mu_A$ for the corresponding protocol: $\mu_A = t$ for BB84 [58] and $\mu_A = 2\sqrt{t}$ for SARG protocol [59]. Using

this in Eq (2) voa we get

$$\alpha_{\rm A} = 10 \log \mu_{\rm B} \text{ dB (for BB84)},$$

$$\alpha_{\rm A} = 10 \log \mu_{\rm B} + 5 \log t - 10 \log 2 \text{ dB (for SARG)}. \tag{3}$$

The value of $\alpha_{\rm VOA}$ can be calculated from Eq (1) as

$$\alpha_{\rm VOA} = 9.2 \text{ dB (BB84)},$$

$$\alpha_{\rm VOA} = 7.7 + 2.5 \log t \text{ dB (SARG)}. \tag{4}$$

Thus, to set the optimal value of $\mu_A$, for the BB84 protocol the system sets the VOA to a constant value while for the SARG protocol the value for the attenuation depends on the channel loss, and varies in a range roughly between 1.7 and 7.2 dB for the distances of a typical QKD link using SARG (120 to 10 km respectively).

In order to quantify the performance of the attack we need to estimate the extra mean photon number per pulse $\mu_{\rm ext}$ coming out of Alice due to the injection of light from Eve. Let's assume that Eve sends external light that reaches the fiber spool and manages to couple a mean photon number $\mu_{\rm inj}$ inside the fiber in a pulse towards the PM that passes the PM during its phase-modulation window. These photons go through the PM, reflect from the FM, then pass through the PM, the delay line, the VOA and the coupler to come out to the channel. The total loss experienced by these photons is

$$\alpha_{\rm T} = 2\alpha_{\rm PM} + \alpha_{\rm spool} + \alpha_{\rm VOA} + \alpha_{\rm coup}. \tag{5}$$

For BB84, this becomes $\alpha_{\rm T} = 33$ dB while for SARG, this becomes $\alpha_{\rm T} = 31.5 + 2.5 \log t$ dB. The extra mean photon number coming out of Alice is then

$$\mu_{\rm ext} = \mu_{\rm inj} 10^{-\alpha_{\rm T}/10}. \tag{6}$$

## 3.3 Experiment

In order to determine the mean number of injected photons $\mu_{\rm inj}$, we used a c.w. laser at 1536 nm with a power of 17.2 mW. We sent a collimated laser beam through the ventilation hole into the delay-line fiber spool (Fig 3). A single-photon detector (ID Quantique ID201) with efficiency $\eta_{\rm d} = 0.1$, detection gate width $T = 20$ ns, and gate repetition rate $f_{\rm s} = 100$ kHz was used to measure the injected light. The selected gate width is close to the time the phase modulator is active ($\sim 20$ ns [20]), thus we can directly use our photon count without any time normalization. The gate rate was chosen in order to allow sufficient time between the gates to avoid afterpulsing. The photon count data was collected from measurement at the output of the optical fiber spool. The measured dark count rate was $N_{\rm dc} = 25.7$ counts per second and the average count rate with the c.w. laser on was $N = 58.95$ counts per second for the best case (maximum coupling), with both averages given from an integration time of 20 s. We maximized the number of coupled photons with a two-step procedure. First we identified the best path for the light through the metal grid into the fiber spool, finding the spots on the surface of the spool for which the photon count was higher. By varying the height of the input point and the direction of the beam, we selected the best entry points for the attack. The second optimization stage was done by fine-tuning the angle of the beam.

Assuming a Poissonian photon number distribution, we can write

$$1 - e^{-\eta_{\rm d}\mu_{\rm inj}} = \frac{N - N_{\rm dc}}{f_{\rm s}}. \tag{7}$$

This gives the mean photon number per pulse (or per gate) $\mu_{\text{inj}} = 3.32(21) \times 10^{-3}$. Using Eqs (4) to (6), the corresponding value of $\mu_{\text{ext}}$ are found to be

$$\begin{aligned} \mu_{\text{ext}} &= 3.32 \times 10^{-6.3} = 1.56 \times 10^{-6}, &\text{(BB84)} \\ \mu_{\text{ext}} &= 3.32 \times 10^{-6.15} t^{-0.25} = 2.35 \times 10^{-6} t^{-0.25}. &\text{(SARG)} \end{aligned} \qquad (8)$$

## 3.4 Attack analysis

We will show the effects of the light injection attack by studying the maximum key rate between Alice and Bob. The maximum key rate of a QKD system (in bits per pulse) is the highest number of bits Alice and Bob can extract from the photon exchange and privacy amplification stages and still be confident that the eavesdropper has no relevant information about the resulting bits. Alice and Bob can create a secret key if their mutual information $I(A:B)$ is greater than the mutual information between Alice and Eve $I(A:E)$. The optimal key rate is $R_{\text{opt}} = \max_{A' \leftarrow A}\{I(A':B) - I(A':E)\}$ where the optimization is in $A'$, the result of local processing at Alice's side.

We will extrapolate our experimental results to different laser powers in a scenario where Alice does no post-processing on her measured bits [$R = I(A:B) - I(E:B)$]. Our reference is the mean $\mu_{\text{inj}} = 3.32 \times 10^{-3}$ photons per gate for 17.2 mW, which corresponds to an external illumination energy of 0.34 nJ in each time bin. While the 17.2 mW laser cannot inject enough light to compromise the system, if Eve uses a stronger laser, the results are quite different. We show examples that correspond to realistically achievable laser energies in the range of 4 − 10 μJ for each 20 ns data pulse, assuming the number of photons scales linearly with the laser power. The photons Eve manages to sneak into the system at these power levels can significantly reduce the maximum key rate. Alice and Bob, believing the outgoing photon number is smaller, might create insecure keys.

Clavis2 uses two protocols: BB84 for short distances, for channel loss up to 3 dB, and SARG04 for longer links up to channel loss of around 20 dB. While a QKD system may be designed to work at a higher loss, in Clavis2 technical limitations due to issues such as synchronization restrict the maximum channel loss to 20 dB, according to ID Quantique (private communication, 2018). For high laser powers and long distance links, Alice and Bob underestimate how much information Eve can learn and use an insecure key rate.

For SARG04 we use an approximate key rate formula [59] that is valid against different incoherent attacks (including photon number splitting) in the limit where the interference visibility $V = 1$. The key rate is [Eq (90) in Ref. [59]]

$$R \approx \frac{\eta}{4}[1 - I_S(1)]\left(\mu t - \frac{\mu^3}{12}\right). \qquad (9)$$

Here $I_S(1) = 1 - H\left(\frac{1}{2} + \frac{1}{2}\sqrt{1 - \frac{1}{2}}\right)$ is Eve's information if she has access to a quantum memory (storage attack), with $H(x) = -x \log_2(x) - (1-x)\log_2(1-x)$ being the binary entropy function. The actual mean photon number coming out of Alice is $\mu' = \mu + \mu_{\text{ext}}$. Using Eq (8) we can write

$$\mu' = 2\sqrt{t} + \mu_{\text{inj}} 10^{-3.15} t^{-0.25}. \qquad (10)$$

The key rates under the attack are calculated by replacing $\mu$ with $\mu'$ in Eq (9) for different injected photon values $\mu_{\text{inj}}$, which are extrapolated from our experiment assuming a linear growth with laser power.

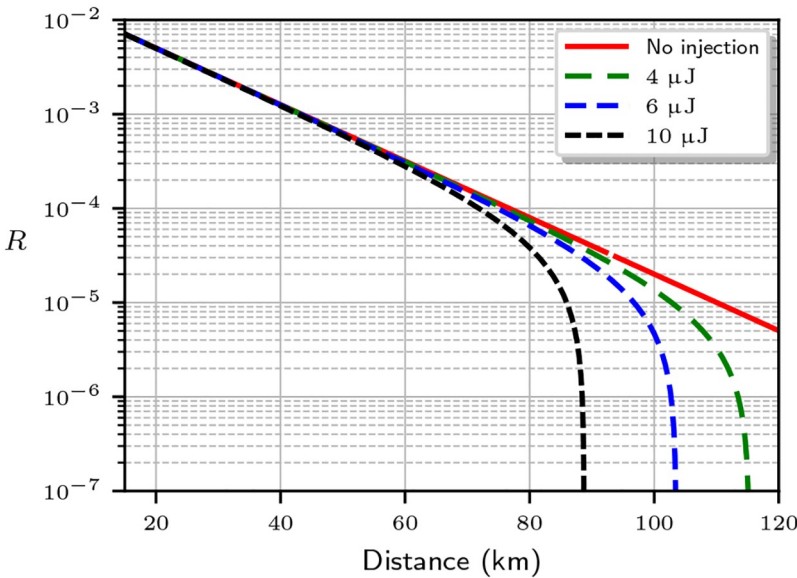

**Fig 4. Evolution of the secret key rate R for SARG04 with the link distance for an attenuation of 0.2 dB/km.** The red line shows the rate limit estimation Alice and Bob make with their available data. The dashed lines show the corrected rate limit when the mean photon number includes the injected photons from an attacker using lasers with different total energies in the 20 ns data pulses. The maximum distance for a secure key becomes smaller at higher laser power.

Fig 4 shows the expected key rate assuming an optimal mean photon number $\mu_A = 2\sqrt{t}$ (red solid line), and compares it to the actual key rate limit taking into account the increase in the mean photon number at the output for different laser powers (dashed lines). For the simulation, we used an optical fiber attenuation of 0.2 dB/km. We see that the maximum distance for secure communication drops for higher powers. An abrupt cutoff appears as $I(A:E) > I(A:B)$, making secret key extraction impossible. For those powers and distances, Eve can compromise the key generation process with the injected photons.

We model the attacks on BB84 assuming combined photon-number-splitting and cloning attacks [58]. From the mutual information [Eqs (22) and (26) in Ref. [58]], we get a key rate

$$R = \frac{1}{2}(\mu t \eta + 2 p_d)[1 - H(Q)] - \frac{1}{2}\mu t \eta\left[\left(t - \frac{\mu}{2}\right)I_1(D_1) + \frac{\mu}{2}\right] \qquad (11)$$

for a quantum bit error rate (QBER)

$$Q = \frac{1}{2} - \frac{V}{2\left(1 + \frac{2p_d}{\mu t \eta}\right)}, \qquad (12)$$

in a system using a detector with efficiency $\eta$ and a dark count probability per gate $p_d$. $I_1(D_1) = 1 - H\left(\frac{1}{2} + \sqrt{D_1(1 - D_1)}\right)$ with $D_1 = \frac{1-V}{2 - \mu/t}$.

Fig 5 shows the key rate limit for a detector with efficiency $\eta = 0.1$, dark count probability $p_d = 5 \times 10^{-5}$, and visibility $V = 0.99$. The red (solid) line shows the key rate estimation for the optimal mean photon number $\mu = t$, which is compared to the key rates when the mean photon number is $\mu' = \mu + \mu_{ext}$, with $\mu_{ext}$ given by Eq (8). Note that under attack, the secure key rate $R$ slightly improves for most transmission distances. This is because $\mu = t$ we are using is a commonly accepted approximation. A true optimal photon number that maximizes $R$ is slightly

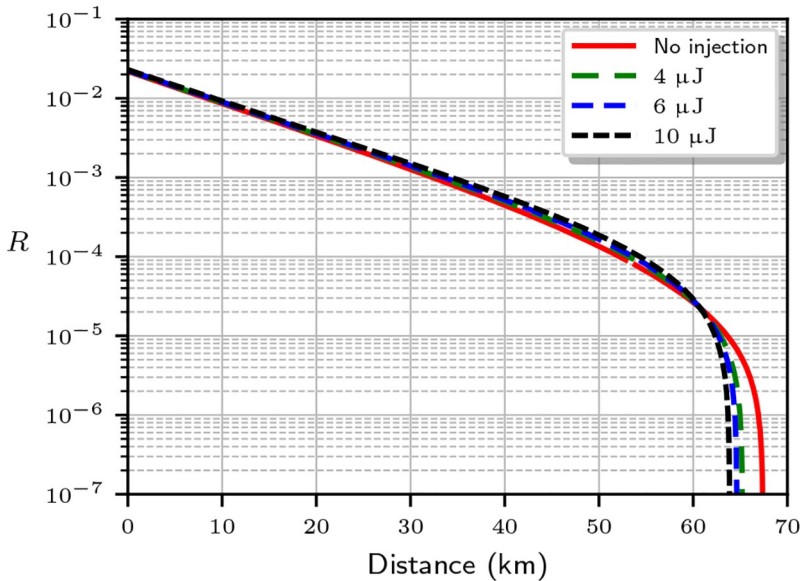

**Fig 5. Evolution of the secret key rate $R$ for BB84 with the link distance for an attenuation of 0.2 dB/km.** The red line shows the rate limit estimation Alice and Bob make with their available data. The dashed lines show the corrected rate limit when the mean photon number includes the injected photons from an attacker using lasers with different total energies in the 20 ns data pulses. The maximum distance for a secure key becomes smaller at higher laser power. We assumed a detector efficiency $\eta = 0.1$ with dark count probability $p_d = 5 \times 10^{-5}$ and visibility $V = 0.99$.

different and can be obtained by a numerical optimization. We have verified that the latter produces similar plots, except that the attack then always reduces $R$.

To summarise, in Clavis2 the SARG protocol may be compromised with lasers producing an energy of $\sim 4$ μJ in 20 ns or less for links in the 60–100 km range. For BB84, the injected photons have a negligible effect on the key rate for the short-distance links where Clavis2 uses that protocol. However a system using BB84 for long-distance transmission might also be vulnerable to light injection attacks. It has been shown that even a small unexpected increase of the output mean photon number may break the security of decoy-state BB84 and measurement-device-independent QKD protocols [60], which are often used at longer distances. While the known combined photon-number-splitting and cloning attacks [58] require a quantum memory, the difference between the assumed and the actual key rates opens a loophole in the security of the system and must be fixed. Otherwise we cannot claim physical security where the only limitation on the attacker is what is allowed by the laws of quantum mechanics. A QKD system should be safe not only against present attacks, but also against future, more technologically advanced attackers (who may have a quantum memory).

For a c.w. laser, 4 μJ in 20 ns translates into 200 W, which, while achievable, requires specialized lasers and could have negative side effects. At such high powers the laser could damage the plastic cover of the fiber spool, affect other components with the reflected light, or even injure people coming close the attacked equipment. However, pulsed lasers can provide the necessary energy per pulse under realistic scenarios.

In order to choose the best parameters for the pulsed laser, we need to study the communication protocol implemented in Clavis2. We take as a reference the data structure for our unit [23], where the data pulses are sent grouped in packets called "frames" with a period of 1 ms and each frame contains 1000 data pulses of 20 ns length with 200 ns period. Most of the time there is no transmission, but if we want to inject photons into each data pulse, we need a

pulsed laser with a repetition rate of 5 MHz. Each laser pulse must be no longer than 20 ns and have an energy in the microjoule range. If the attacker injects her pulses into the system perfectly, she only needs to send an average power in the range of a few watt. For 4 μJ per pulse and a total of $10^6$ pulses per second, the average power for the attack is only 4 W.

At 1550 nm the eye is less sensitive to damage and, while these power levels are not safe, they are not extreme. Similarly, previous laser damage experiments have shown that a few watt can damage detectors when applied directly but not most of the parts in a QKD setup [26, 61]. Extrapolating these c.w. results to the possible damage to the exposed fiber or the covering plastic, it seems that light injection attacks at a few watt may be successful and remain undetected.

We can find many commercial lasers close to the needs of the attack: the 1550 nm lasers used in ranging applications such as self-driving cars are in the right range of pulse energies, repetition rates and pulse lengths [62].

Pulsed lasers present additional complications. Coupling to the fiber does not happen at a single strand of fiber in the spool. Light coupling is distributed at different points. While Eve could shape her pulses to maximize the energy that gets into the time bin where the PM is active, a realistic attack with a pulsed laser will not manage to inject the full pulse energy into the 20 ns bin of modulation.

There are two limit situations. In the worst case, Eve is limited to c.w. lasers, like in our experiment. In the best case there is a single point of insertion and pulsed lasers in the range of a few watt are sufficient. The single point of insertion might still be a reasonable assumption. We have attacked the fiber spool because it offers the largest target visible from the outside, but we have observed that light directed to fiber connectors or unprotected pigtails also couples inside the fiber. If there is a line of sight to these single points of insertion, attacks become viable in the lower power range.

In a realistic attack, Eve will likely be in between these two extremes. Even with distributed coupling in the spool, she can probably reduce the time her laser is active to about the frame length and thus reduce her average power to a few tens of watts.

If Eve is far from the QKD device, beam divergence owing to atmospheric turbulence and diffraction could also pose a problem. However the pulsed lasers we have suggested for the attack [62] come from ranging applications and they are already prepared to cover large distances. Most of them work close to the diffraction limit (with a beam quality factor $M^2$ between 1.1 and 1.4). In any case, an attacker should try to work as close to the device as possible.

Taking all these details into consideration, we can compare the light injection attack to the existing Trojan-horse attacks [55, 56]. In the latter, Eve co-opts the public optical channel between Alice and Bob to send light probes so that she can learn the configuration of the different optical components on each side, particularly the state of the phase modulator.

There are two important differences. First, the legitimate channel is an essential part of the communication between Alice and Bob and must always be present. However, it is an expected point of entry and there are multiple countermeasures like detectors to monitor the input and filters to attenuate unwanted wavelengths [23, 26]. A light injection attack uses unsuspected paths into the fiber for which there are no planned countermeasures. Once the photons are inside the fiber, they go together with the legitimate signal undetected. The main purpose of this paper is to raise awareness of this vector of attack so that these paths are blocked during the design of the system.

Second, for the light injection path we have found in our device, light coupling is quite inefficient. Trojan-horse light at 1550 nm can be attenuated between 60 to 110 dB, with different values at other wavelengths depending on the deployed countermeasures [57, 63, 64]. In our experiment, 17.2 mW resulted in an average of $3.32 \times 10^{-3}$ photons per 20 ns bin ($2.13 \times 10^{-11}$

mW at 1550 nm, giving a total attenuation of 119 dB before all the losses in Alice, including the variable attenuation stage). In principle, the Trojan-horse attacks seem easier to launch, as they require less power, but they are easier to detect because the input point is known.

## 4 Attack on a quantum random number generator

Measures against light injection attacks should also be considered when building other quantum optical devices. A clear example are quantum random number generators. Many existing commercial and lab QRNGs work with different quantum states of light [29], making the coupling of external light a potential security problem.

We present a proof-of-concept attack on a prototype of a quantum random number generator [65] with an internal LED that can be seen from a ventilation hole. The same path allows a way to the detectors inside the box and a flashlight directed at certain angles can bias the generated bit sequence.

Fig 6 shows the scheme of the random number generator we have investigated. Photons from the LED go through a 50:50 beam splitter (BS) and have an equal probability of going out each of its two outputs. Each output leads to a photon detector (two photomultiplier tubes PMT1 and PMT2). Photocounts on PMT1 toggle an electrical signal from a low to a high logic level. Photocounts on PMT2 toggle the transition from the high to the low logic level. If the electrical signal is already in the low (high) state, there is no effect. The resulting signal is then sampled at regular intervals to produce a random bit (0 for the low level, 1 for the high level). If the photon detection rate is sufficiently higher than the sampling rate, the resulting bit sequence will be close to random. The photons are produced using a regular LED in front of a sealed metal box with a pinhole leading to the beam splitter and the input to the detectors.

The current through the LED can be adjusted to modify the count rate. For count rates of the order of tens of MHz, the average time between consecutive photons is much greater than the coherence time of the source and we can ignore the effects of interference [65]. The resulting electrical signal is sampled at a rate of 1 MHz.

Most of the light from the LED does not reach the detectors. In order to reach the single-photon level after the pinhole, the LED emits at a classical power level. In fact, the LED generates light in visible wavelengths and can be seen from the outside.

The whole setup of the studied QRNG was inside a nuclear instrumentation module (NIM) that went into a NIM crate. The module enclosure, made of metal, had four groups of ventilation holes [Fig 6(b)]. The most accessible path for our attack was through the holes at the top of the module close to the front panel (shown as a blue line in the picture).

Three units of the prototype were used in research setups. We did most of our work on two of them and obtained similar results. There are small variations between the units and they have their own control software, very similar to what can be expected from a commercial device.

Our attack works by flooding the legitimate photons from the LED with a much stronger beam coming from the outside. We produced the latter with a handheld LED flashlight (Mini-Maglite AA). The injected light reaches the pinhole and, once inside the box and depending on the angle, it is reflected and scattered in multiple directions. By manually varying the input angle, we could maximize the amount of light going into PMT2. The increased photon number in just one detector biases the sequence in favor of 1s in the device we did most of the work with (in other units of the prototype we found the detectors associated with 1 and 0 were swapped).

This rudimentary attack is enough to show that it is possible to bias the output sequence. We used the control software to store two sequences of around 15 MB, one generated under

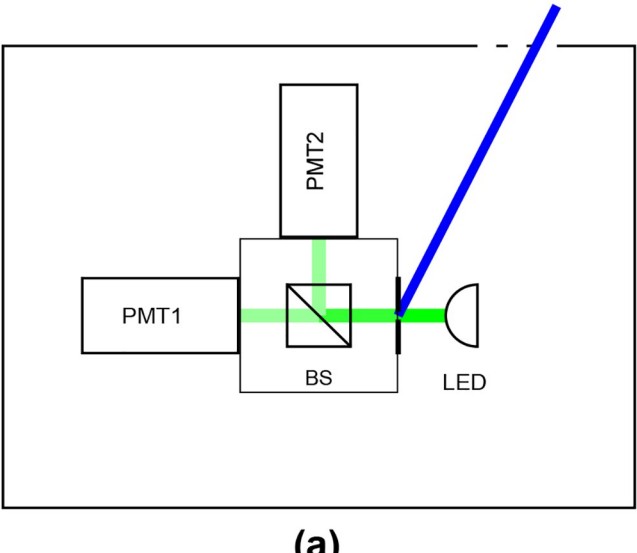

**(a)**

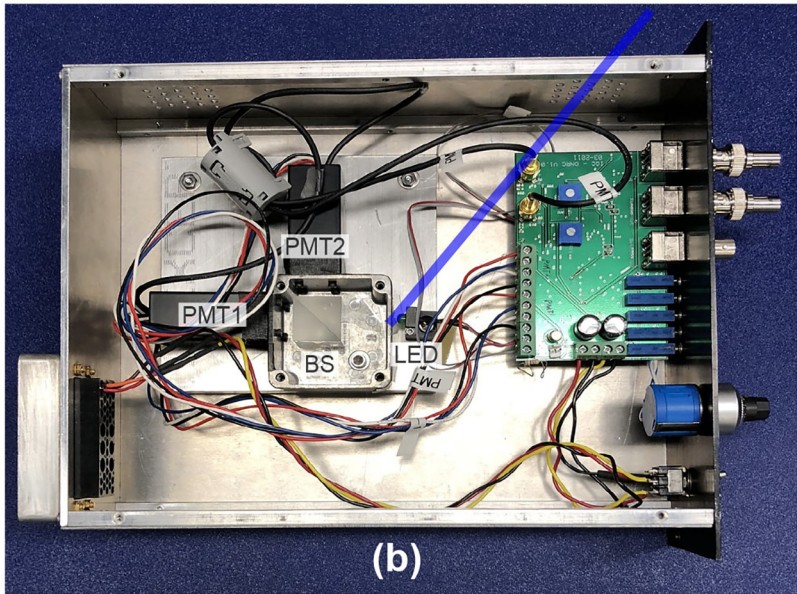

**(b)**

**Fig 6. Quantum random number generator under attack.** (a) Scheme of the QRNG. Light from a LED (green beam) passes a pinhole and a beam splitter (BS) with two outputs leading to detectors PMT1 and PMT2. For our attack we take advantage of a ventilation hole at the top of the case. With additional light, we can make one detector more likely to click. We show in blue a possible path from the ventilation hole to the pinhole that gives access to the metal box with the BS. (b) Picture of the prototype QRNG [65], with covers removed from the enclosure and beam splitter box. The path to the pinhole has been marked with a blue line.

normal operation and one generated while directing the output of the flashlight to the ventilation hole. We managed to obtain a ratio > 80% for the number of ones in the final generated sequence just by moving the flashlight into an adequate angle (99200493 bits were 1 while 22503955 bits were 0). Using the Linux utilities `ent` [66] and `rngtest` [67] we could also see that, while under normal operation the results of the $\chi^2$-test [68] and the FIPS-140-2 tests [46] were consistent with a uniform random sequence, the bits generated during the light injection

attack failed these tests. The sample files are included in the S1 Data, together with the results of applying the different randomness tests to the sequences.

This is more than enough to launch a successful attack and reduce the entropy of the output in this particular QRNG design. Even small biases in the random sequences can compromise many cryptographic protocols [69]. For instance, in QKD, if the random numbers in the basis and bit selection are biased, the protocol becomes insecure [70].

A light injection attack is not limited to biasing the bit sequence towards one of the values. At the beginning of the random bit generation, there is a calibration stage during which the PMT voltages are adjusted to optimize detection [65]. An attacker could introduce an intermediate level of light for the calibrated state (50% generation rate for 0s and 1s). By removing the light Eve can increase the probability of getting 0. By increasing the light, she can bias the bit towards 1. The light could be invisible for the people in the room. While we used white light, the detectors' efficiency peaks at UV wavelengths and an attacker could hide the injected light from the operators of the QRNG.

The device we have tested is only a prototype. More advanced QRNGs should include some real-time monitoring and debiasing. It is recommended that physical random number generators include internal systems that check for operation errors [71–74]. However, our attack could be refined to use pulsed lasers to circumvent these countermeasures. The best solution is thus to eliminate any light coupling path.

## 5 Discussion and recommendations

We have shown that ventilation openings can be a problem for the security of quantum key distribution systems and quantum random number generators. If they are not properly protected, an external attacker can use them to couple light inside the optical part of the device.

Light coming from the wrong direction in the normal QKD optical channel can also alter the intended operation of the system. For instance, external laser light can alter the photon sources in QKD devices and has been shown to weaken security either by seeding the source laser so that consecutive pulses are not phase-independent [75], altering the wavelength to identify the state choice [76, 77], increasing the mean photon number [23, 60, 77], or performing laser machining to physically alter the components inside the QKD setup [26, 61]. In this paper, we have focused on light entering from unsuspected places, such as ventilation openings.

In our experiment, we have targeted the delay line in a plug-and-play QKD system, where a long spool of optical fiber allows for a large coupling area. Similarly, we have been able to bias the output of a prototype for a quantum random number generator. There are other quantum devices with exposed delay lines or detectors that could be vulnerable to similar attacks. Portions of unshielded fiber or uncovered pigtails and connectors might also offer a way inside the optical part of the quantum device.

Precaution suggests any optical component should be hidden from external light. Our light injection attack requires a line of sight with the device, but servers close to a window could be targeted from the outside of the building (see Fig 1). While in our QKD unit we have targeted the largest component, the fiber spool, further experiments with other components show how photons can sometimes be injected at pigtails, even with higher efficiency. The solution is as simple as covering all the sensitive parts with an opaque material. An explicit design against light injection should be considered when building QKD devices.

At the moment of writing, light injection attacks are not explicitly taken into account when designing QKD systems. While some commercial QKD systems are completely enclosed [78], the enclosure seems to be designed in compliance with security standards against physical

probing and electromagnetic emissions, and not to avoid light injection attacks. ID Quantique states that proper countermeasures against the latter type of attack have been implemented in their current generation of QKD products [44].

There is a growing effort in the standardization of QKD [79–87] and the existing drafts already include provisions for ventilation holes with respect to physical probing, following the example of previous secure device standards. For instance, the ETSI GS QKD 008 Group Specification [86] includes the same requirement as the NIST's FIPS 140-2 standard [46]. Both ask that

> "If the cryptographic module contains ventilation holes or slits, then the holes or slits shall be constructed in a manner that prevents undetected physical probing inside the enclosure (e.g. require at least one 90 degree bend or obstruction with a substantial blocking material)."

Similarly, there are provisions against an attacker learning the internal configuration of the device from direct visual observation of the ventilation holes or slits, as stated in the ETSI GS QKD 008 Group Specification [86]:

> "If the QKD module contains ventilation holes or slits, then the holes or slits shall be constructed in manner to prevent the gathering of information of the module's internal construction or components by direct visual observation using artificial light sources in the visual spectrum. . ."

While these physical probing attacks also seem unlikely, it is important to consider them in the standards. The risk can be greatly reduced with little effort during the design stage. The light injection attacks belong to this group of potential problems. We believe standards for QKD should include a similar requirement to prevent them, for instance by asking that

> "If the QKD module contains ventilation holes or slits, then the holes or slits shall be constructed in a manner such that there is no direct line of sight to the optical components inside the enclosure. The device shall also be built so that any indirect path to the optical components, either by reflection or by diffuse scattering inside the device, is sufficiently attenuated so that external light with power up to hundreds of watts reaching the ventilation holes or slits is unable to inject enough photons into the optical scheme to compromise the security of operation. Alternatively, the optical components shall be enclosed in a separate section without holes. Alternatively, each individual optical component and their connections shall be completely covered with an opaque material."

## Supporting information

**S1 Data.**
(ZIP)

## Acknowledgments

The authors thank ID Quantique for cooperation, technical assistance, and providing the QKD hardware. We thank Thomas Jennewein for providing the QRNG prototype, Chris Pugh and Jean-Philippe Bourgoin for their assistance with it.

## Author Contributions

**Conceptualization:** Juan Carlos Garcia-Escartin, Vadim Makarov.

**Formal analysis:** Juan Carlos Garcia-Escartin, Shihan Sajeed.

**Funding acquisition:** Vadim Makarov.

**Investigation:** Juan Carlos Garcia-Escartin, Shihan Sajeed.

**Project administration:** Vadim Makarov.

**Resources:** Vadim Makarov.

**Supervision:** Vadim Makarov.

**Validation:** Juan Carlos Garcia-Escartin, Shihan Sajeed.

**Visualization:** Vadim Makarov.

**Writing – original draft:** Juan Carlos Garcia-Escartin, Shihan Sajeed.

**Writing – review & editing:** Juan Carlos Garcia-Escartin, Shihan Sajeed, Vadim Makarov.

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
