## [Decision Letter · Decision Letter 0]

18 Dec 2019

PONE-D-19-29328

Attacking quantum key distribution by light injection via ventilation openings

PLOS ONE

Dear Dr. Garcia-Escartin,

Thank you for submitting your manuscript to PLOS ONE. After careful consideration, we feel that it has merit but does not fully meet PLOS ONE’s publication criteria as it currently stands. Therefore, we invite you to submit a revised version of the manuscript that addresses the points raised during the review process.

If you decide to submit a revision, please be sure that you have addressed all of the detailed comments of the reviewers below. The issues raised can be addressed with some careful rewriting, and the addition of appropriate analysis and details. Repeating experiments is not necessary.

If you are planning to submit a revision, please let us know.

We would appreciate receiving your revised manuscript by Jan 31 2020 11:59PM. To enhance the reproducibility of your results, we recommend that if applicable you deposit your laboratory protocols in protocols.io, where a protocol can be assigned its own identifier (DOI) such that it can be cited independently in the future. For instructions see: http://journals.plos.org/plosone/s/submission-guidelines#loc-laboratory-protocols

We look forward to receiving your revised manuscript.

Kind regards,

Gorjan Alagic, Ph.D.

Academic Editor

PLOS ONE

Reviewers' comments:

Reviewer's Responses to Questions

**Comments to the Author**

1. Is the manuscript technically sound, and do the data support the conclusions?

Reviewer #1: Partly

Reviewer #2: No

2. Has the statistical analysis been performed appropriately and rigorously? 

Reviewer #1: Yes

Reviewer #2: No

3. Have the authors made all data underlying the findings in their manuscript fully available?

Reviewer #1: No

Reviewer #2: No

4. Is the manuscript presented in an intelligible fashion and written in standard English?

Reviewer #1: Yes

Reviewer #2: Yes

5. Review Comments to the Author

Reviewer #1: 1. Is the manuscript technically sound, and do the data support the conclusions?

The manuscript is technically sound, however, the reason why I think the data does not go all the way to support their conclusions is in the attached PDF report.

3. Have the authors made all data underlying the findings in their manuscript fully available?

I could not find if this has been done but I am fairly sure that this is merely a formality they either forgot or it was may be not clear that it was required at this stage.

Reviewer #2: In this article, the authors exploit ventilation holes in quantum devices to inject light inside it and either get information or modify the behaviour of the device itself.

In the first part of the article, the authors study a Trojan horse attack performed by injecting light directly into the SMF delay line of a commercial plug-and-play QKD system, thus preventing it from being detected by the photodiode placed at the entrance of the device. In the second part, they attack a simple QRNG prototype with the aim of inducing an imbalance in the detection rate of the two detectors, thus modifying the output statistics.

The idea behind the article is interesting, since it addresses some implementation issues that might completely break the security of the device. However, in my opinion, there are some major points, in the experimental evaluation of the attacks, that make the article, as it is now, unsuitable for publication.

To evaluate the effect of light injection on QKD, the authors shine a collimated laser into the ventilation holes of the Alice part of Clavis 2 QKD system, aiming at the fiber spool used as a delay line, and look at the number of photons just after the spool. By measuring the fraction of photons coupled into the fiber, they get the injection efficiency as a function of the input laser power and they estimate, from the losses of the different components in a typical QKD setting, the number of photons injected into the line going to Bob. An attacker can thus increase the number of photons going from Alice to Bob without being detected. The main problems in this part are:

* there is no description of the properties of the beam used in the injection. Since this article deals with attacks on the implementation, it should contain more information on how this attack can be implemented practically.

* the analysis of the attack considers just the effect of an undetected increase of the mean photon number coming out from Alice. What would change if Eve used a light at a slightly different wavelength, putting a wave-division multiplexer in the channel collecting all the light she injected without affecting the one going to Bob?

* the authors base the analysis on the attack model described in ref [57]. One of the hypothesis in their analysis is that Eve measures her quantum system just after the sifting procedure. This requires her to have a quantum memory, something that is not realistic nowadays.

* the authors show their attack to be effective for laser powers going from 200 to 500 W. While it is true that these powers are realistically achievable, it is not clear that they don't give rise to other issues (like the degradation of the components). I think the authors should be more careful about this point, since it is crucial in their work.

The following section, about the attacks through ventilation holes on a QRNG, is even more problematic. Indeed, the authors just say that injecting light into the QRNG it is possible to bias the statistics of the two detectors, something that is well known. The prototype QRNG they show in the photo is almost 20 years old, so it has nothing to do with current technology in QRNGs. Moreover, there are no data supporting the claims made by the authors (except a generic "we managed to obtaina ratio >80% ...", with no description of the setup used for obtaining it). In order to get something meaningful, the authors should at least compare the statistics obtained with and without the attack, showing explicitly which advantage could an adversary gain with it.

As a summary, the authors address a real security issue in the security of quantum devices, i.e., the accessibility of optical components through ventilation holes, but they don't provide enough evidence on how this could be exploited, now, by an eavesdropper. I think that, before being considered again for publication, the authors should address more carefully all the experimental aspects behind the proposed attacks, focusing on what is feasible with current technology.

6. PLOS authors have the option to publish the peer review history of their article (what does this mean?). If published, this will include your full peer review and any attached files.

Reviewer #1: Yes: Nitin Jain

Reviewer #2: No

---

## [Author Response · Author response to Decision Letter 0]

5 Jun 2020

The response has been uploaded in a pdf file.

---

## [Decision Letter · Decision Letter 1]

13 Jul 2020

Attacking quantum key distribution by light injection via ventilation openings

PONE-D-19-29328R1

Dear Dr. Garcia-Escartin,

We’re pleased to inform you that your manuscript has been judged scientifically suitable for publication and will be formally accepted for publication once it meets all outstanding technical requirements.

Kind regards,

Kim-Kwang Raymond Choo, Ph.D.

Academic Editor

PLOS ONE

Additional Editor Comments (optional):

Reviewers' comments:

Reviewer's Responses to Questions

**Comments to the Author**

1. If the authors have adequately addressed your comments raised in a previous round of review and you feel that this manuscript is now acceptable for publication, you may indicate that here to bypass the “Comments to the Author” section, enter your conflict of interest statement in the “Confidential to Editor” section, and submit your "Accept" recommendation.

Reviewer #1: All comments have been addressed

Reviewer #2: All comments have been addressed

2. Is the manuscript technically sound, and do the data support the conclusions?

Reviewer #1: Yes

Reviewer #2: Yes

3. Has the statistical analysis been performed appropriately and rigorously? 

Reviewer #1: Yes

Reviewer #2: Yes

4. Have the authors made all data underlying the findings in their manuscript fully available?

Reviewer #1: Yes

Reviewer #2: Yes

5. Is the manuscript presented in an intelligible fashion and written in standard English?

Reviewer #1: Yes

Reviewer #2: Yes

6. Review Comments to the Author

Reviewer #1: The authors have addressed all concerns and revised the manuscript appropriately. I am happy to recommend it for a publication.

Reviewer #2: In their resubmitted version of the article, the authors have answered in a satisfactory way to the remarks of both reviewers. As it is now, the article is suitable for publication.

I have just one small advice: in the answer to the second point of reviewer 2, the authors claim that, "given the low coupling we found in our device, Eve’s best strategy seems to be using also the photons in the legitimate channel". I think that this observation could be included in the main article, since it explains explicitly why the authors decided to study attacks where the injected photons have the same wavelength as the photons used for QKD. However, this is really a minor issue and the authors could decide to ignore it.

7. PLOS authors have the option to publish the peer review history of their article (what does this mean?). If published, this will include your full peer review and any attached files.

Reviewer #1: **Yes: **Nitin Jain

Reviewer #2: No

---

## [Editor Report · Acceptance letter]

23 Jul 2020

PONE-D-19-29328R1 

Attacking quantum key distribution by light injection via ventilation openings 

Dear Dr. Garcia-Escartin:

I'm pleased to inform you that your manuscript has been deemed suitable for publication in PLOS ONE. Congratulations! Your manuscript is now with our production department. 

Kind regards, 

on behalf of

Cloud Technology Endowed Professor Kim-Kwang Raymond Choo 

Academic Editor

PLOS ONE